# Heavy metal distribution and ecological risk in surface sediments of the Bohai Sea

**Shilin Li**[1,2,3], **Jianlei Chen**[2,3], **Xuzhi Zhang**[2,3], **Jianshe Zhang**[1], **Yongjiang Xu**[2,3]*,
**Yong Xu**[2,3]*

**1** National Engineering Research Center For Marine Aquaculture, Zhejiang Ocean University, Zhoushan, Zhejiang, China, **2** National Key Laboratory of Mariculture Biobreeding and Sustainable Goods, Yellow Sea Fisheries Research Institute, Chinese Academy of Fishery Sciences, Qingdao, China, **3** Yellow Sea Fisheries Research Institute, Chinese Academy of Fishery Sciences, Qingdao, China

* xuyj@ysfri.ac.cn (YX); xuyong@ysfri.ac.cn (YX)

## Abstract

Heavy metal contamination in marine sediments poses significant ecological risks, particularly in semi-enclosed seas like the Bohai Sea, where limited water exchange exacerbates pollution retention. Heavy metals are persistent, bioaccumulative, and toxic, making their assessment crucial for environmental management. This study investigated the spatial distribution, seasonal dynamics, and potential ecological risks of heavy metal contamination in the central Bohai Sea, with an emphasis on regulatory interventions and anthropogenic influences. The annual average concentrations of Cu, Zn, Pb, Cd, Hg and As in surface sediments were 15.951, 32.556, 15.234, 0.250, 0.028 and 2.628 mg/kg, respectively, all below China's Class I Marine Sediment Quality Standards. Seasonal variations revealed peak concentrations in August for Zn, Pb, Hg and As, likely driven by increased terrestrial inputs and hydrodynamic conditions. Cd exhibited the highest ecological risk, with a single-factor risk index exceeding 30 in May, followed by Hg, Pb, Cu, As and Zn. The comprehensive pollution index remained below 5 across all seasons, indicating overall low pollution levels. However, localized exceedances of Class I standards for Cu, Pb and Cd were observed, particularly in summer and autumn. Spatially, metal concentrations were higher near industrial and riverine discharge zones, with anthropogenic sources such as petrochemical industries, aquaculture, and urban runoff contributing significantly. This study highlighted seasonal and spatial heterogeneity in heavy metal contamination in the central Bohai Sea, emphasizing the influence of industrial activities and hydrodynamic processes. While overall pollution levels were low, the high ecological risk associated with Cd underscores the need for continued monitoring and targeted pollution control measures. Strengthening enforcement of industrial regulations, improving sediment management, and addressing seasonal fluctuations in pollutant inputs were critical for mitigating future risks. These findings provided a scientific

**Data availability statement:** All relevant data are within the manuscript and its Supporting Information files.

**Funding:** This study was supported by the Central Public-interest Scientific Institution Basal Research Fund, CAFS (NO.2023TD53). The funders had no role in the study design, data collection and analysis, decision to publish, or preparation of the manuscript.

**Competing interests:** The authors have declared that no competing interests exist.

foundation for sustainable marine environmental management and policy formulation in the Bohai Sea.

## 1 Introduction

Heavy metal pollution has becomed a severe issue facing global marine ecosystems, particularly in semi-enclosed waters where the accumulation of these toxic elements can have long-lasting impacts on aquatic environments. Heavy metals are highly toxic, persistent, resistant to degradation, and bioaccumulative, allowing them to transfer through the food chain, impacting marine organisms and ultimately jeopardize human health [1]. In seawater, heavy metals can interact with suspended particles through adsorption, complexation, and precipitation, leading to their accumulation in surface sediments [2,3]. Influenced by tides, waves, and human activities, heavy metals stored in sediments can be released back into the overlying seawater, resulting in "secondary pollution" [4] This dynamic transformation between "source" and "sink" poses a serious threat to marine ecosystems [5,6].Consequently, sediments play a critical role in assessing heavy metal pollution due to their dual functions of releasing and accumulating these contaminants [7].

Sediments enriched with heavy metals can pose potential risks to the overlying water through various chemical and biological processes [8,9], making it crucial to accurately assess the quality and ecological risk of marine sediments in this region [10] Numerous studies have shown that metal-contaminated sediments significantly harm marine ecosystem health, with heavy metal bioaccumulation and the amplification of biotoxicity being among the most critical concerns [11,12]. Additionally, the high discharge of anthropogenic pollutants in China's coastal waters has led to deteriorating water quality, increased occurrences of red tides, damage to fishery resources, and frequent pollution incidents, placing considerable ecological pressure on coastal areas [13,14] With the rapid economic development of the Bohai Economic Rim, the Bohai Sea is facing immense pressure from environmental pollution and ecological degradation linked to human activities. The region's primary economic activities include commerce, fisheries, electronics, petrochemicals, metal smelting, automobile manufacturing, biotechnology and modern medicine, alkali production, food processing, textiles, and building materials [12,15,16]. Studies indicate that heavy metal pollution in Bohai Bay is primarily attributed to wastewater from these industries.

In recent years, numerous marine researchers have conducted extensive studies on heavy metals in the Bohai Rim region [17]. The concentrations of heavy metals in surface sediments are relatively low in Laizhou Bay, but higher along the northwestern coast of the Bohai Sea, decreasing from the nearshore areas to the open sea [18,19]. Notably, no contamination of Cr, Cu or Zn was detected in Laizhou Bay, while As, Cd and Pb exhibited mild to moderate pollution levels [20,21]. Due to their high toxicity,Cd and As pose significant potential ecological risks. Risk areas generally diminish with increasing distance from the shore, indicating that heavy metal

pollution and associated risks are primarily influenced by anthropogenic activities [9,18,22].In the coastal areas of Liaodong Bay, heavy metals concentrations in surface sediments follow the order of Pb > Zn > Cu > Cd, attributed to increase riverine input after the rainy season. The pollution levels are relatively higher in autumn and summer, and lower in spring and winter [23]. In the Changli ecological monitoring area of northwestern Bohai Bay, heavy metals in sediments pose moderate to severe risks, with Cd identified as the primary contributor to ecological risk [24]. Most areas of the Bohai Sea, northern estuaries, the Yellow River estuary, and the nearshore areas of the Liaodong Peninsula, show moderate Cd and Hg pollution [21]. Nickel and chromium in the sediments of the Bohai coastal waters primarily originate from rock sources, while other metals (Zn, As, Cd, Cu, Hg and Pb) mainly derive from anthropogenic discharges [14]. Sediment testing in the spill area of the Penglai 19−3 oilfield revealed that heavy metal concentrations were higher than in nearby regions such as Bohai Bay, Liaodong Bay, and the Shandong Peninsula coast, with pollution levels increasing in the order of Zn < Cd < Cu < Pb < Ni < Cr [17]. Both As and Hg concentrations exceeded the Class I sediment quality standards [1]. Existing studies have primarily focused on pollution conditions along the Bohai coast and specific areas (such as the coastlines of Bohai Bay, Liaodong Bay, Laizhou Bay and river estuaries), while research on heavy metal pollution in surface sediments of the central Bohai region remains relatively scarce. Previous studies on sediment pollution in the Bohai Sea typically involved one or two sampling events per year, making it difficult to comprehensively reveal the seasonal variation patterns of heavy metals in sediments, especially in the central Bohai region, where systematic data on year-round changes are still lacking. Furthermore, the 2011 oil spill from the Penglai 19−3 oilfield, which allowed crude oil to migrate through geological faults and coarse sediments before reaching fine sediments, has become a significant source of heavy metal pollution in the central Bohai Sea [1,17].

The global nature of this issue is further evidenced by studies in regions such as the Mediterranean Sea, where historical contamination from industrial activities has led to elevated concentrations of metals such as cadmium and mercury [25]. Similarly, the ecological health of marine environments in Southeast Asia has been increasingly impacted by heavy metals due to rapid industrialization and agricultural runoff [14].

Due to emerging oil exploration, industrial development and oil refineries, the heavy metal content−particularly Hg and Cd−has been rising along the entire Persian Gulf. This increase is primarily attributed to traffic emissions, industrial activities and the dredging of aquatic sediments [26,27].Meanwhile, an analysis of trace element concentrations in sediment cores from the Red Sea-Gulf of Aqaba coast and the southern Gulf of Mexico revealed high concentrations of As, Cr, Cu, Pb and Zn.These findings likely reflect the impact of active natural oil seeps and strong river drainage, leaving traces in the sedimentary record akin to oil extraction activities [28].

The Penglai 19−3 oilfield is situated in the south-central region of the Bohai Sea (120°01′E–120°08′E, 38°17′N–38°27′N),with an average water depth ranging from 27 to 33 meters. A significant oil spill event occurred on June 4, 2011 resulted in substantial contamination of seabed sediments surrounding the Penglai 19−3 oilfield and extending northwestward. The contaminated sediments covered an area of approximately 1,600 km$^2$,exceeding China's Class I marine sediment quality standards, with a severely impacted core zone of 20 km$^2$ surpassing Class III standards.

The Bohai Sea is a semi-closed marginal sea in China,connecting to the Yellow Sea through the Bohai Strait. It̕s limited water exchange capacity results in poor self-purification ability, making ecological recovery difficult if severe pollution occurs. As a result, the central Bohai Sea is designated as a key area for pollution processes and ecological risks, playing a crucial role in the comprehensive management of environmental issues.

This study involved four cruise samplings missions in the central Bohai Sea in 2013, conducted in May, August, October and December.Approximately 150 sediment samples were collected to determine the concentrations of heavy metals (Cu, Zn, Pb, Cd, Hg and As) in the sediments. We analyzed the spatial distribution, seasonal variations, and pollution levels to assess the heavy metal contamination in the central Bohai Sea. The aims of this study is to provide data support that support an understanding of the ecological risk mechanisms of heavy metals in marine surface sediments and to offer a scientific basis for future environmental management and pollution control strategies in the region.

## 2 Materials and methods

### 2.1 Study area

The Bohai Sea comprises Laizhou Bay, Bohai Bay and Liaodong Bay, with the remaining area referred to as the central Bohai Sea. This semi-enclosed shallow water region was connected to the Yellow Sea via the Bohai Strait, covering a surface area of approximately 70 000 km², with a coastline length of about 2,700 km, and an average depth of around 18 m [29]. The coastal region was home to several major cities, including Beijing, Tianjin, Tangshan, Qinhuangdao and Dalian, along with diverse activities such as industry, aquaculture, tourism and shipping, all of which contributed to significant pollution pressure on the Bohai Sea [30]. The research area was located in the central Bohai Sea, China (38.21°N–39.01°N; 118.98°E–120.83°E), with 41 sampling stations established (S1 Table and Fig 1). Sampling was conducted in May, August, October and December of 2013, representing spring, summer, autumn and winter, respectively.

### 2.2 Sample collection and processing

Sediment samples were collected using a grab sampler, and the undisturbed surface layer (0−2 cm) was scooped with a plastic spoon and sealed in polyethylene bags, and stored at 0−4°C for subsequent analysis. In the laboratory, the sediment samples were thawed to room temperature, dried in an oven at 80°C for 24 hours, and then ground thoroughly with an agate mortar. The samples were sieved through a 180 μm standard mesh and after thorough mixing, subsamples were taken for analysis. Sediments were digested using microwave-assisted digestion [29]. The concentrations of heavy metals in the sediments were determined according to *The Speciation for Marine Monitoring−Part 5: Sediment Analysis* (GB 17378−2007) [30]. In 2013, Pb and Cd were analyzed using graphite furnace atomic absorption spectrophotometry, Cu and Zn were measured with flame atomic absorption spectrophotometry, Hg was assessed using cold vapor atomic absorption, and As was determined via atomic fluorescence spectrometry.

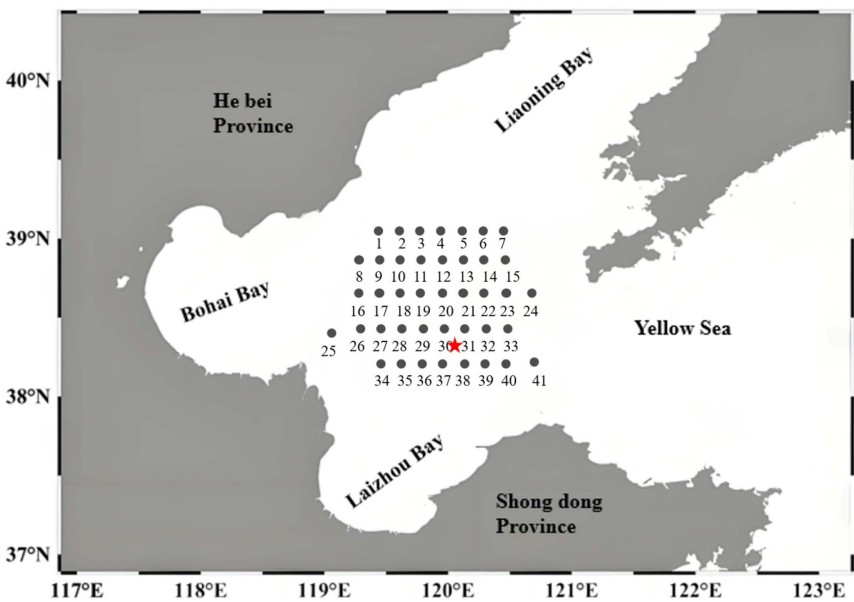

**Fig 1. Surveyed area and sampling locations of surface sediments(The red star mark is the oil spill point of Penglai 19−3).**

## 2.3 Quality assurance and quality control

This study adhered strictly to quality assurance and quality control (QA/QC) procedures, implementing quality control messures through the use of duplicate samples and method blanks during the experiments. The standard sediment reference materials (GBW-07314), provided by the Second Institute of Oceanography, SOA, China, were used to assess the experimental accuracy and precision [31]. It was also utilized for internal quality control. The relative deviations among three parallel reference samples remained within acceptable ranges, with variability in the analysis of standard materials maintained within ±10%. The detection limits for heavy metals in sediments were as follows: Cu 0.04 mg/kg, Cd 0.2 mg/kg, Pb and Zn 0.1 mg/kg, and Hg 0.002 mg/kg. The recoveries for Cu, Pb, Zn, Cd, As and Hg were 98.2-103.1%, 97.8-102.8%, 98.7-103.2%, 98.3-102.1%, 98.1-102.7%, and 98.4-102.6%, respectively. Glassware used in the experiments was pre-cleaned by soaking in 15% nitric acid (w/w) for at least 24 hours, followed by rinsing with deionized water. All reagents used were of analytical grade or higher.

## 2.4 Statistical analysis

Geographic information was visualized using Golden Software Surfer to ensure precise mapping and accurate representation of spatial relationships. Data preprocessing and validation were conducted using Microsoft Excel 2021 to guarantee the accuracy and completeness of the datasets. Subsequently, bar charts were generated using OriginPro 2020b to enhance clarity and accuracy in data presentation.

## 2.5 Potential ecological risk assessment method of sediments

Numerous methods were employed both domestically and internationally to assess heavy metal pollution in sediments, including sediment quality standards, the geo-accumulation index, potential ecological risk index, fingerprinting methods, and biological effects databases. The most widely used method is the potential ecological risk index developed by Swedish scientist Hakanson [32]. In this study, we applied this method to evaluate the ecological risk of heavy metals in the sediments of the central Bohai Sea, with the calculation formula as follows: (1) and (2).

$$E_r^i = T_r^i \times C_f^i \qquad (1)$$

$$RI = \sum_i^n E_r^i = \sum_i^n \left( T_r^i \times C_f^i \right) = \sum_i^n \left( \frac{T_r^i \times C_s^i}{C_n^i} \right) \qquad (2)$$

In the formula: $RI$ represents the potential ecological risk index for all heavy metals, while $E_r^i$ is the potential ecological risk factor for metal i. $T_r^i$ is the toxic response factor of heavy metals, reflecting their toxicity levels and the sensitivity of organisms to heavy metal pollution. The values are as follows: Cu = 5, Zn = 1, Pb = 5, Cd = 30, Hg = 40 and As = 10. The enrichment factor of heavy metals $C_f^i$ is calculated as $C_f^i = C_s^i / C_n^i$, where $C_s^i$ represents the measured concentration of heavy metals in surface sediments, and $C_n^i$ is the background value. In this study, the pre-industrial background values of heavy metals were used (Table 1). The specific classification of potential ecological risk levels for heavy metals is shown in Table 2.

**Table 1. The background reference values of heavy metals.**

| Region | Cu | Zn | Pb | Cd | Hg | As | Resource |
|---|---|---|---|---|---|---|---|
| Pre-industrialization | 30 | 80 | 25 | 0.5 | 0.2 | 15 | [32] |

**Table 2. The relation between evaluation indexes and the contamination degree and potential ecological risk.**

| $C_f^i$ | Single factor pollution level | $C_d$ | Comprehe sive pollution level | $E_r^i$ | Single factor ecological risk | $E_{RI}$ | Comprehensive potential ecological risk |
|---|---|---|---|---|---|---|---|
| <1 | Low | <5 | Low | <40 | Low | <150 | Low |
| 1~3 | Moderate | 5~10 | Moderate | 40~80 | Moderate | 150~300 | Moderate |
| 3~6 | High | 10~20 | High | 80~160 | High | 300~600 | High |
| ≥6 | Serious | ≥20 | Serious | 160~320 | Serious | ≥600 | Serious |
| | | | | ≥320 | Very serious | | |

## 3. Result

### 3.1 Heavy metals content in surface sediments

The concentrations of heavy metals in surface sediments from the central Bohai Sea were presented in S2 Table and Table 3 and Fig 2. Overall, the levels of heavy metals (Cu, Zn, Pb, Cd, Hg, As) in the surface sediments were relatively low, with annual averages of 15.951, 32.556, 15.234, 0.250, 0.028 and 2.628 mg/kg respectively, all below the Class I standards for Marine Sediment Quality. In May,the average concentrations of Cu, Zn, Pb, Cd, Hg and As were 17.49, 31.29, 12.86, 0.58, 0.005 and 0.230 mg/kg, respectively. At some sampling sites, the concentrations of Cu, Pb and Cd exceeded the Class I standards, with exceedance rates of 2.94%, 2.94% and 14.71%, respectively, while Zn, Hg and As remained below the standards. In August, the average concentrations were 20.27, 46.55, 18.08, 0.15, 0.066 and 6.515 mg/kg. Some sampling sites showed Hg concentrations exceeding the standards, with an exceedance rate of 5%, while the other elements remained below the Class I standards. The October data indicated average concentrations of Cu, Zn, Pb, Cd, Hg and As at 12.98, 25.52, 17.50, 0.16, 0.008 and 1.931 mg/kg, respectively. Some sites had Cd concentrations exceeding the standards, with an exceedance rate of 2.78%, while the other elements did not exceed the standards. In December,average concentrations of Cu, Zn, Pb, Cd, Hg and As were 13.76, 28.42, 12.87, 0.14, 0.034 and 2.088 mg/kg, respectively, with all elements remaining below the Class I standards.

### 3.2 Seasonal variations of heavy metals in surface sediments of the central Bohai Sea

The results of the seasonal study indicate significant fluctuations in the concentration of Cu in May from 2.99 to 106.99 mg/kg, with an average of 17.49 mg/kg.This variability may be related to hydrodynamic conditions and biological activities in early summer. In August, the concentration range narrowed 6.76 to 34.64 mg/kg, but the average concentration increased to 20.27 mg/kg, suggesting a possible accumulation of Cu within the sediments. By October and December, the concentration range further narrowed to 5.27-30.08 mg/kg and 2.17-25.75 mg/kg, and averages of 12.98 mg/kg and 13.76 mg/kg, respectively, indicating that Cu deposition and desorption processes stabilized during the autumn and winter seasons.

**Table 3. The content of the heavy metal in surface sediments in central region of Bohai sea.**

| Month | Exceedance | Cu(mg/kg) | Zn(mg/kg) | Pb(mg/kg) | Cd(mg/kg) | Hg(mg/kg) | As(mg/kg) |
|---|---|---|---|---|---|---|---|
| May | Exceedance rate | 2.94 | 0.00 | 2.94 | 14.71 | 0.00 | 0.00 |
| Aug | Exceedance rate | 0.00 | 0.00 | 0.00 | 0.00 | 5.00 | 0.00 |
| October | Exceedance rate | 0.00 | 0.00 | 0.00 | 2.78 | 0.00 | 0.00 |
| December | Exceedance rate | 0.00 | 0.00 | 0.00 | 0.00 | 0.00 | 0.00 |
| Annual | Range | 2.17-106.99 | 5.92-61.19 | 0.68-76.76 | 0.02-4.98 | 0.001-0.34 | 0.011-13.994 |
| | Avg | 16.123 | 32.556 | 15.234 | 0.250 | 0.028 | 2.628 |
| | SD | 3.392 | 9.369 | 2.855 | 0.216 | 0.028 | 2.685 |
| | Class I standard for sediments | 35 | 150 | 60 | 0.5 | 0.2 | 20 |

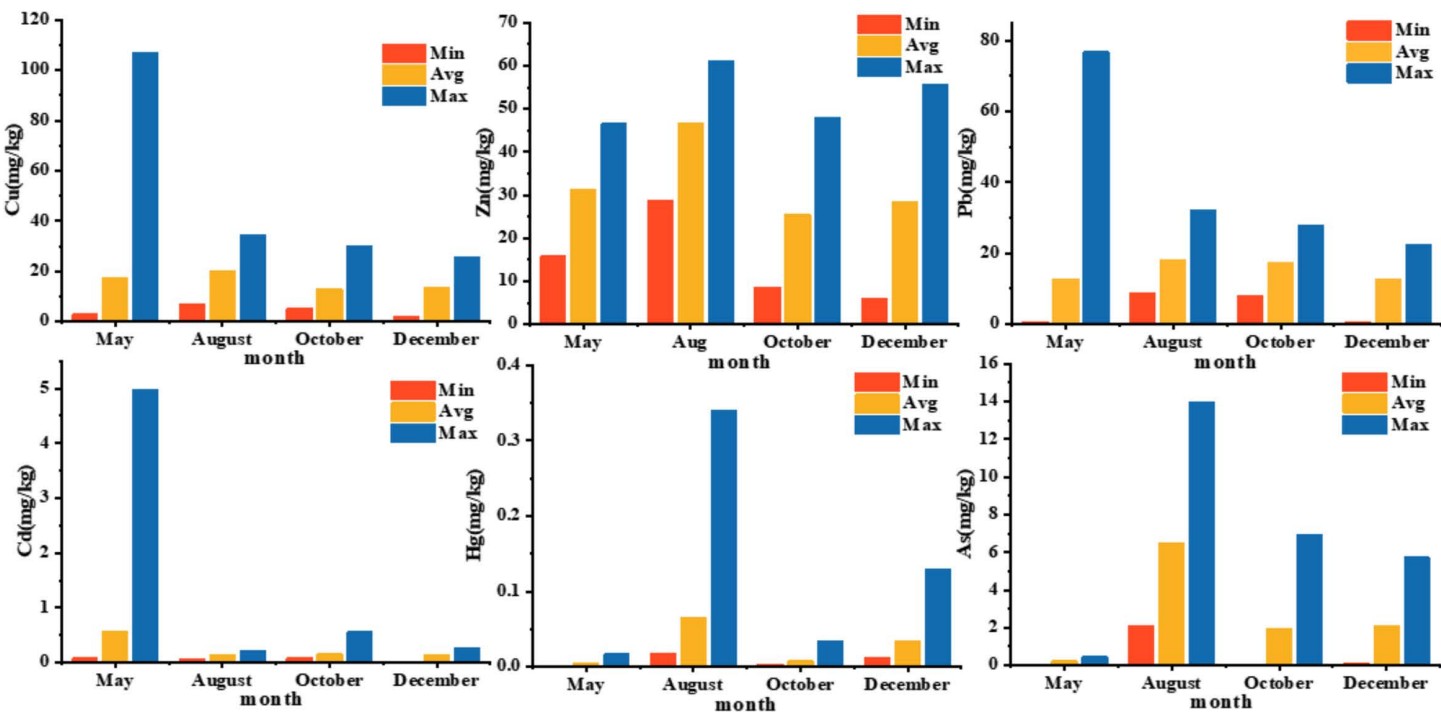

**Fig 2. Seasonal extremes and averages for various metals.**

For Zn, the concentration in May ranged from 15.84 to 46.54 mg/kg, with an average of 31.29 mg/kg, indicating relatively active sedimentation during spring. In August, the range significantly increased to 28.77-61.19 mg/kg, with an average of 46.55 mg/kg, suggesting that Zn accumulation during the summer may have been influenced by increased terrestrial inputs. By October and December, Zn concentrations decreased to 8.61-47.92 mg/kg and 5.92-55.80 mg/kg, with averages of 25.52 mg/kg and 28.42 mg/kg, respectively, indicating Zn reduced inputs and deposition during the autumn and winter seasons.

The concentration of Pb in May exhibited a wide range (0.7-76.76 mg/kg, with an average of 12.86 mg/kg), reflecting diverse and fluctuating sources of Pb in spring. In August, the range narrowed to 8.79-32.29 mg/kg, with the average increasing to 18.08 mg/kg, suggesting more concentrated sources and increased Pb input during summer. In October and December, Pb concentrations further decreased, ranging from 8.22-27.90 mg/kg and 0.68-22.60 mg/kg, with averages of 17.50 mg/kg and 12.87 mg/kg, respectively, indicating a gradual reduction in Pb deposition during the autumn and winter seasons.

The concentration of Cd in May ranged from 0.08 to 4.98 mg/kg, with an average of 0.58 mg/kg, indicating higher sedimentation activity during early summer. In August, Cd concentrations significantly decreased to 0.06-0.22 mg/kg, with an average of 0.15 mg/kg, suggesting weakened Cd deposition during summer. In October and December, Cd concentrations ranged from 0.09-0.56 mg/kg and 0.02-0.27 mg/kg, with averages of 0.16 mg/kg and 0.14 mg/kg, respectively, indicating that stabilization of Cd deposition during autumn and winter.

Hg concentrations in May were relatively low (0.001-0.017 mg/kg, with an average of 0.005 mg/kg), indicating low Hg input during this period. In August, the range expanded to 0.017-0.340 mg/kg, with the average rising to 0.066 mg/kg, suggesting increased Hg accumulation during summer. In October and December, Hg concentrations ranged from

0.003-0.034 mg/kg and 0.013-0.129 mg/kg, with averages of 0.008 mg/kg and 0.034 mg/kg, respectively, indicating relatively stable Hg deposition during autumn and winter.

In May, the concentrations of As ranged from 0.011 to 0.462 mg/kg, with an average of 0.230 mg/kg, indicating significant As deposition during early summer. In August, the range of As concentrations increased significantly to 2.083-13.994 mg/kg, with an average of 6.515 mg/kg, possibly associated with enhanced surface runoff due to summer precipitation. By October and December, As concentration ranges narrowed to 0.044-6.899 mg/kg and 0.095-5.733 mg/kg, with averages of 1.931 mg/kg and 2.088 mg/kg, respectively, indicating a decrease in As deposition activity during the autumn and winter seasons.

### 3.3  Analysis of heavy metal pollution levels in surface sediments of the central Bohai Sea

The results presented in Table 4 indicate that the single factor pollution index for Cd in May exceeded 1, signifying moderate pollution. Conversely,the single factor pollution indexs for Cu, Zn, Pb, Hg and As in May were all below 1, indicating low pollution levels. Throughout August, October and December, the single factor pollution indices for all six heavy metals (Cd, Zn, Pb, Cu, Hg and As) remained under 1. The comprehensive pollution indices for heavy metals in all months were below 5, indicating a low level of pollution(S3 Table and Fig 3). In 2013,the average single factor pollution coefficients exhibited a decreasing trend in the following order: Pb, Cu, Cd, Zn, As and Hg, all of which were less than 1, indicating low pollution level.

In terms of temporal distribution, the single factor pollution indices for Cu and Zn were ranked as follows: August > May > December > October. For Pb, the order was: August > October > May > December. The single factor pollution index for Cd was highest in May, followed by October, August and December. For Hg and As, the order was: August > December > October > May. This variability in pollution indices reflected the differing seasonal influences on heavy metals.The comprehensive pollution index for heavy metals changed as follows: August > May > October > December, with August showing the highest overall pollution level. This pattern suggested that the hydrological conditions of coastal runoff during high-flow and low-flow periods were the primary factors influencing the seasonal variation of heavy metal pollution in the sediments of the central Bohai Sea (S4 Table and Fig 4).

### 3.4  Potential ecological risk assessment of heavy metals in surface sediments of the central Bohai Sea

This study evaluated the potential ecological risks of heavy metals in the surface sediments of the central Bohai Sea, focusing on two key indicators: the potential ecological risk coefficient and the risk index for heavy metals (Table 5).

Analyzing the potential ecological risk coefficients across different months revealed that the values for all six heavy metals were below 40, with average values ranging from 0.41 to 15.36. The average values,listed in descending order, Cd, Hg, Pb, Cu, As and Zn. Notably, Cd exhibited the highest potential ecological risk coefficient in the study area, while Zn had the lowest(S5 Table and Fig 5). This variation may be attributed to the differences in the nature of terrigenous

Table 4. The results of single factor evalution on heavy metal pollution in sediment.

| Month | Single factor pollution coefficient.($C_f^i$) | | | | | | Comprehensive pollution coefficient($C_d$) |
|---|---|---|---|---|---|---|---|
| | Cu | Zn | Pb | Cd | Hg | As | |
| May | 0.58 | 0.39 | 0.51 | 1.16 | 0.03 | 0.02 | 2.69 |
| Aug | 0.68 | 0.58 | 0.72 | 0.30 | 0.33 | 0.43 | 3.04 |
| October | 0.43 | 0.32 | 0.70 | 0.32 | 0.04 | 0.13 | 1.94 |
| December | 0.46 | 0.36 | 0.51 | 0.27 | 0.17 | 0.14 | 1.91 |
| Avg | 0.54 | 0.41 | 0.61 | 0.51 | 0.14 | 0.18 | 2.40 |
| SD | 0.115 | 0.115 | 0.116 | 0.432 | 0.140 | 0.175 | 0.561 |

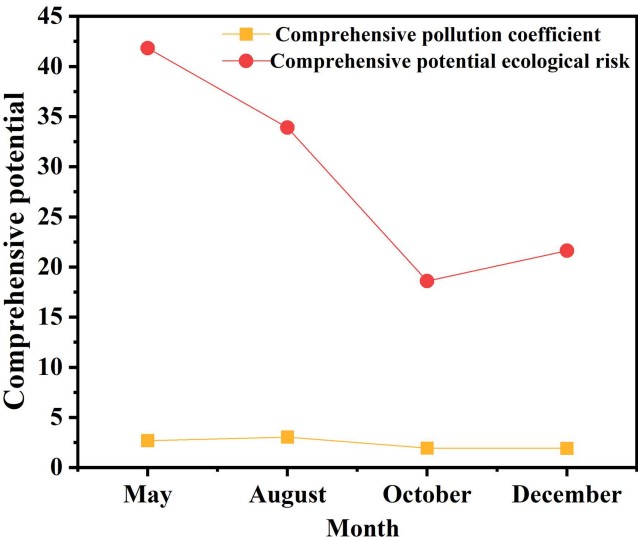

**Fig 3. Comprehensive pollution of heavy metals in surface sediments in the central Bohai Sea.**

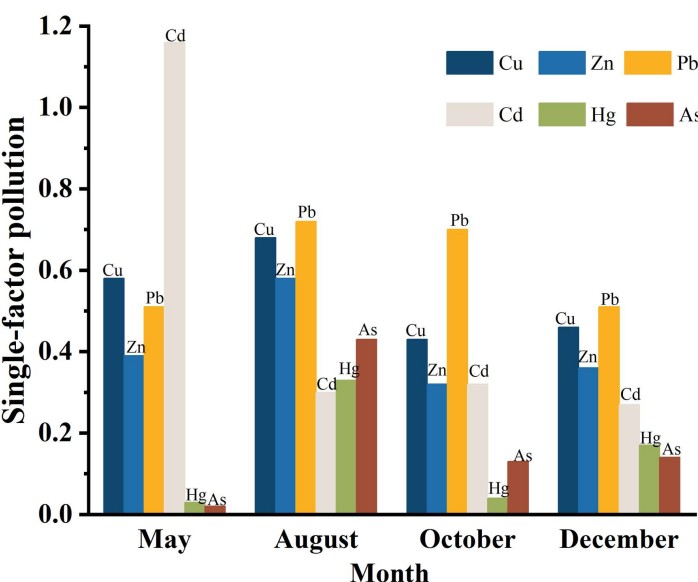

**Fig 4. Single factor evaluation of heavy metal pollution in sediments.**

pollutants entering the area and the biogeochemical cycling of heavy metals within the sedimentary environment. When examining the comprehensive potential ecological risk indices across months, all values remained below 150, indicating a low potential ecological risk level. The index was highest in May and lowest in October. Further analysis of the single factor ecological risk indices for heavy metals showed the following trends: for Cu and Zn, the order from highest to lowest was August＞May＞December＞October. For Pb,the order was August＞October＞December＞May. For Cd, the rankings were: May＞October＞August＞December. For Hg and As,the order was: August＞December＞October＞May.

**Table 5. The potential ecological risk factors ($E_r^i$) and risk indices($E_{RI}$) of heavy metals in surface sediments.**

| Month | Single factor ecological risk | | | | | | Comprehensive potential ecological risk |
|---|---|---|---|---|---|---|---|
| | Cu | Zn | Pb | Cd | Hg | As | |
| May | 2.91 | 0.39 | 2.572 | 34.78 | 1.01 | 0.15 | 41.82 |
| Aug | 3.38 | 0.58 | 3.617 | 8.87 | 13.12 | 4.34 | 33.91 |
| October | 2.16 | 0.32 | 3.500 | 9.65 | 1.67 | 1.29 | 18.59 |
| December | 2.29 | 0.36 | 2.573 | 8.12 | 6.89 | 1.39 | 21.63 |
| Avg | 2.69 | 0.41 | 3.065 | 15.36 | 5.67 | 1.79 | 28.99 |
| SD | 0.567 | 0.115 | 0.571 | 12.965 | 5.619 | 1.789 | 10.819 |

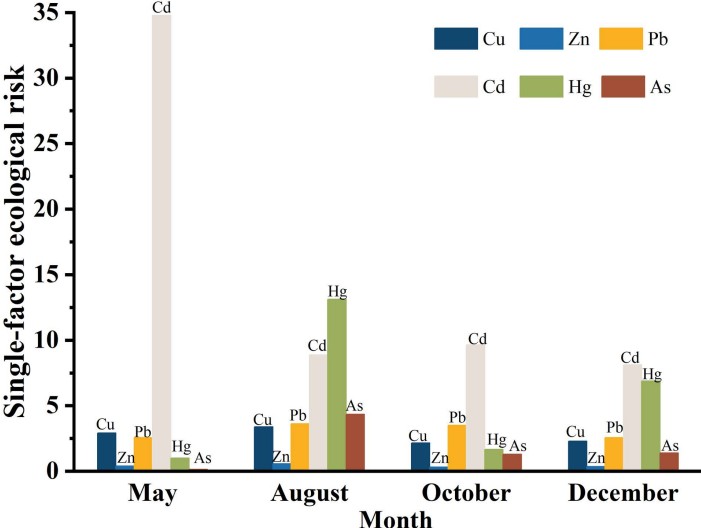

**Fig 5. The potential ecological risk factors and risk indices of heavy metals in surface sediments.**

The overall comprehensive heavy metal pollution index for different months, ranked from highest to lowest was: May＞August＞December＞October, with May reflecting the highest overall potential ecological risk(Fig 3).

## 4. Discussion

### 4.1. Seasonal variation of heavy metals concentrations in the central Bohai Sea

Seasonal variations in heavy metal concentrations in the central Bohai Sea revealed distinct patterns (S6 Table and Fig 6). The concentrations of Cu and Zn followed the order: summer＞spring＞winter＞autumn. Previous studies (Liu et al.,2016), identified the summer sequence of metal concentrations as Zn＞Cu＞Pb＞Cd, while Zhang et al. (2017) repoted significant seasonal variations in Pb, Zn, and As, with the highest concentrations in winter, Cu and Hg peaking in spring, and Cd in autumn.Liu et al (2023) found that in the Bohai Sea,average metal concentrations during summer followed the order: Zn＞ As＞ Pb＞Cu＞Cd, with As peaking in autumn. In spring and early summer, hydrodynamic processes driven by monsoon winds and increased river discharge intensified (e.g., Yellow River inputs), promoting sediment resuspension and remobilizing Cu adsorbed onto fine particles [13,33].As hydrodynamic energy decreased in late summer, finer sediments enriched with heavy metals settle [8].During the rainy season (June–August), anthropogenic inputs from agricultural runoff and industrial discharges further enriched Cu concentrations [1,34], while summer monsoon rainfall amplified

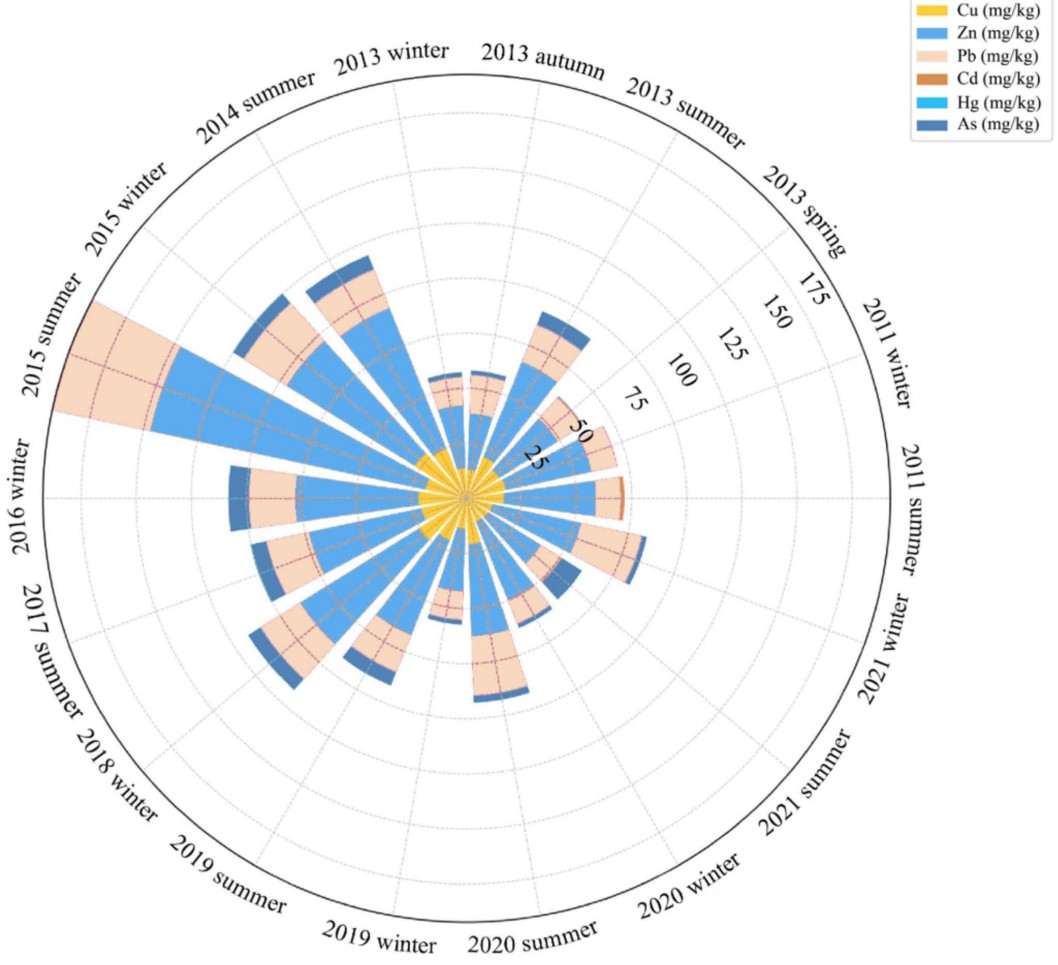

**Fig 6. Seasonal variations of heavy metals in sediments.**

terrestrial inputs such as industrial effluents and coastal aquaculture,These conditions also facilitated the settling of particulate Zn [1,23].

Pb concentrations followed the reasonal trend: summer＞autumn＞winter＞spring. Increased shipping traffic and port activities during summer released Pb containing fuels and antifouling paints, while stagnant water conditions favored sedimentation [34,35].In contrast, reduced coastal erosion and lower industrial activity in colder months helped Pb distribution [14].Cd concentrations followed a different seasonal pattern: spring＞autumn＞summer＞winter.Agricultural runoff, particularly from phosphated fertilizers, and spring algal blooms enhanced Cd adsorption onto organic matter [11,1].

Overall, the total metal concentrations were higher in summer than in autumn, significantly influenced by seasonal variations in temperature and hydrology [36]. From spring (May) to summer (August), rising water temperatures accelerated molecular motion, facilitating the migration and release of heavy metals from sediments into the water column, a phenomenon observed in several Bohai Sea studies [14]. Additionally, reduced pH levels promoted the dissolution of heavy metals bound to carbonates and hydroxides, further increasing their availability in the water phase [13]. The concentrations of Cu, Zn and Pb in surface sediments of the central Bohai Sea were highest during the summer. Hydrological conditions, especially during the high-flow summer period, facilitated the transport of suspended sediments and heavy metal pollutants

into marine areas, leading to notable accumulation in surface sediments.This seasonal release pattern underscored the importance of incorporating hydrological models to predict the movement of contaminants and mitigated their impact on sensitive coastal ecosystems. Researchers such as Zhang *et al.* (2020) advocated for a more integrated approach to sediment management,which included both seasonal monitoring and predictive modeling to assess risks of heavy metal contamination.

Seasonal trends also revealed distinct contamination patterns. Summer maxima for Cu, Zn and Cd (e.g., Zn at 35.2 mg/kg in August 2019 vs. 22.1 mg/kg in winter) were driven by intensified port operations and monsoon-driven sediment remobilization, which amplified metal bioavailability [8,33]. Conversely, winter Pb surges (e.g., 18.4 mg/kg in December 2021 vs. 12.7 mg/kg in July) correlated with coal combustion for heating in coastal cities, releasing Pb-enriched particulates that settle in quiescent winter seas [23].

In conclusion, understanding the seasonal dynamics of heavy metal concentrations in coastal waters like the Bohai Sea is crucial for tracking pollution trends and developing effective management and conservation strategies that protect marine biodiversity. As Liu *et al.* (2023), highlighted, the combination of temperature, pH, and biological activity can create windows of opportunity for intervention, particularly in areas vulnerable to seasonal metal accumulation.

Ecologically, seasonal fluctuations in heavy metal concentrations not only reflect changes in atmospheric or hydrological conditions, but also signal potential shifts in sediment quality and water column health, which can have cascading effects on marine organisms [14,37]. This pose particularly risk to benthic organisms, which rely on sediments for habitat and food. Studies by Ding *et al*. (2019) and Qiao *et al*. (2017) demonstrate that higher temperatures and lower pH levels enhance the mobility of heavy metals, which adversely impacts organisms dependent on sediment for survival.

From a management perspective, these seasonal variations underscore the need to tailor pollution control strategies to temporal fluctuations. For instance, during summer, when metal concentrations are typically higher, more stricter water quality monitorying is essential, particularly in areas with critical aquatic ecosystems such as shellfish farms. This is supported by Chen *et al.* (2020), who emphasize the importance of seasonal assessments of heavy metals to guide decision-making for marine protected areas.

## 4.2 Potential ecological risks of heavy metals in the central Bohai Sea

The ecological risk assessment of heavy metals in Bohai Sea sediments revealed distinct spatial and seasonal variations in contamination patterns. Cadmium (Cd) presented moderate ecological risk ($E_r^i$=34.78), predominantly concentrated in regions adjacent to agricultural runoff zones, aligning with its association with phosphate fertilizer inputs and sediment remobilization [1,13]. Mercury (Hg) exhibited localized elevated risks ($E_r^i$=13.12) in August, likely linked to intensified coastal industrial discharges and hydrodynamic resuspension of legacy pollutants [9,20]. While lead (Pb) and arsenic (As) posed comparatively lower risks, their persistent presence in near-shore sediments underscores the chronic anthropogenic pressures exerted by port activities and untreated urban effluents [21,38]. The overall potential ecological risk index ($E_{RI}$< 150) indicated low-to-moderate risks across the basin, but localized hotspots warrant targeted sediment management to mitigate secondary release during dredging or storm events.

The spatial heterogeneity of heavy metal risks highlights the urgency for adaptive sediment management frameworks. For instance, Cd-enriched zones near the Yellow River delta [1] necessitate stringent controls on agricultural runoff and optimized dredging schedules to reduce resuspension, as sediment disturbance can amplify bioavailable metal fluxes by 20–35% [9,20].Hg hotspots in industrialized bays demand enhanced monitoring of wastewater discharges and stabilization of contaminated sediments using amendments like biochar or calcium silicate [14,39]. Integrating hydrodynamic models with risk indices, as proposed by Yu *et al*. (2019) for the Luanhe Delta, could further predict pollutant dispersal during tidal cycles, enabling proactive containment. Implementing"source-pathway-receptor" interventions—such as phytoremediation in high-risk mudflats [11] —would align with the sediment quality guidelines (GB 17378−2007) and regional ecological restoration targets.

## 4.3 Interannual variations of heavy metals in the central Bohai Sea over the past decade

Comparing sediment metal concentrations across different studies to assess temporal changes in pollution levels is inherently complex due to variations in scale and timing. However, we believe that comparing our finding with historical data on metal concentrations in Bohai Sea sediments is crucial for a deeper understanding of the temporal trends in heavy metal pollution. Consequently, we compared our results with historical data on heavy metals in Bohai Sea sediments [36,40–42] (S7 Table and Table 6 and Fig 7). From 2013 to 2021, the concentrations of Cd, Zn, Pb, Cd, Hg and As in the surface sediments of the central Bohai Sea exhibited significant changes,characterized by interannual fluctuations trends.

Copper (Cu) concentrations exhibited a significant decline from 20.27 mg/kg in 2013 to 11.40 mg/kg in 2021, a trend closely linked to the enforcement of China's Comprehensive Prevention and Control Plan for Heavy Metal Pollution (2018−2021) which imposed stringent industrial emission standards and coastal zone management regulations [39,24]. The elevated baseline in 2013(20.27 mg/kg) likely resulted from sediment-oil interactions remobilizing legacy Cu deposits following the 2011 Penglai 19−3 oil spill, as indicated by isotopic fingerprinting analyses [1,37]. A transient rebound in 2015 (19.17 mg/kg) was plausibly driven by unregulated discharges from shipbreaking operations in Huludao, aligning with environmental compliance reports from regional monitoring stations in Liaodong Bay [41].

Zinc (Zn) concentrations decreased by 32% from 41.87 mg/kg in 2011 to 28.33 mg/kgin 2021, primarily due to the phased decommissioning of electroplating facilities under the Bohai Sea Ecological Restoration Initiative (2016–2020) [16]. However, anomalous peaks in 2015 (68.7 mg/kg) and 2018 (63.49 mg/kg) exceeding regional background levels by 142–187%, correspond with unauthorized effluent discharges from rapidly expanding aquaculture and shipbreaking industries along Liaodong Bay's coastal industrial zones [1,23]. The 2013 baseline (28.42 mg/kg) reflects persistent legacy contamination prior to the implementation of stricter regulatory oversight.

Lead (Pb) exhibited distinct variability, peaking at 26.96 mg/kg in 2020 due to illicit battery manufacturing activities in Liaodong Bay's informal sector. However, concentrations declined sharply to 10.94 mg/kg in 2021 following rigorous enforcement of lead-acid battery recycling regulations under amendments to the Circular Economy Promotion Law

**Table 6. Comparison of heavy metal content in surface sediments of Central Bohai Sea and other sea areas (mg/kg).**

| Central Bohai Sea | Year | Cu (mg/kg) | Zn (mg/kg) | Pb (mg/kg) | Cd (mg/kg) | Hg (mg/kg) | As (mg/kg) | Reference |
|---|---|---|---|---|---|---|---|---|
| Summer | 2011 | 16.84±6.81 | 41.87±11.75 | 11.36±4.25 | 1.65±1.53 | — | — | [43] |
| Winter | 2011 | 18.23±6.17 | 39.74±8.69 | 11.98±4.43 | 0.20±0.12 | — | — | [43] |
| Summer | 2014 | 24.26 | 68.95 | 18.85 | 0.17 | — | 6.89 | [36] |
| Winter | 2015 | 27.61 | 68.7 | 23 | 0.14 | — | 5.73 | [36] |
| Summer | 2015 | 19.17 | 127.66 | 45.11 | 0.39 | — | — | [44] |
| Winter | 2016 | 21.81 | 55.73 | 21.23 | 0.16 | — | 9.21 | [36] |
| Summer | 2017 | 21 | 52.13 | 19.74 | 0.14 | — | 7.35 | [36] |
| Winter | 2018 | 25.56 | 63.49 | 21.21 | 0.15 | — | 6.58 | [36] |
| Summer | 2019 | 20.27±7.22 | 46.55±7.39 | 18.08±5.16 | 0.15±0.05 | 0.06±0.04 | 6.52±2.49 | [40] |
| Winter | 2019 | 13.76±7.75 | 28.42±9.30 | 12.87±6.53 | 0.14±0.07 | 0.03±002 | 2.08±1.43 | [40] |
| Summer | 2020 | 20.67±9.83 | 41.76±4.47 | 26.96±8.03 | 0.22±0.12 | 0.01±0.01 | 3.09±4.26 | [40] |
| Winter | 2020 | 11.20±8.99 | 38.43±7.01 | 11.68±9.25 | 0.18±0.08 | 0.01±0.01 | 1.91±2.66 | [40] |
| Summer | 2021 | 11.40±5.36 | 28.33±8.93 | 10.94±5.11 | 0.38±0.26 | 0.01±0.01 | 10.72±9.76 | [40] |
| Winter | 2021 | 12.09±8.71 | 41.01±3.21 | 28.19±9.52 | 0.27±0.21 | 0.01±0.01 | 2.12±2.91 | [40] |
| Spring | 2013 | 17.49 | 31.29 | 12.86 | 0.58 | 0.005 | 0.230 | This research |
| Summer | 2013 | 20.27 | 46.55 | 18.08 | 0.15 | 0.066 | 6.515 | This research |
| Autumn | 2013 | 12.98 | 25.52 | 17.50 | 0.16 | 0.008 | 1.931 | This research |
| Winter | 2013 | 13.76 | 28.42 | 12.87 | 0.14 | 0.034 | 2.088 | This research |

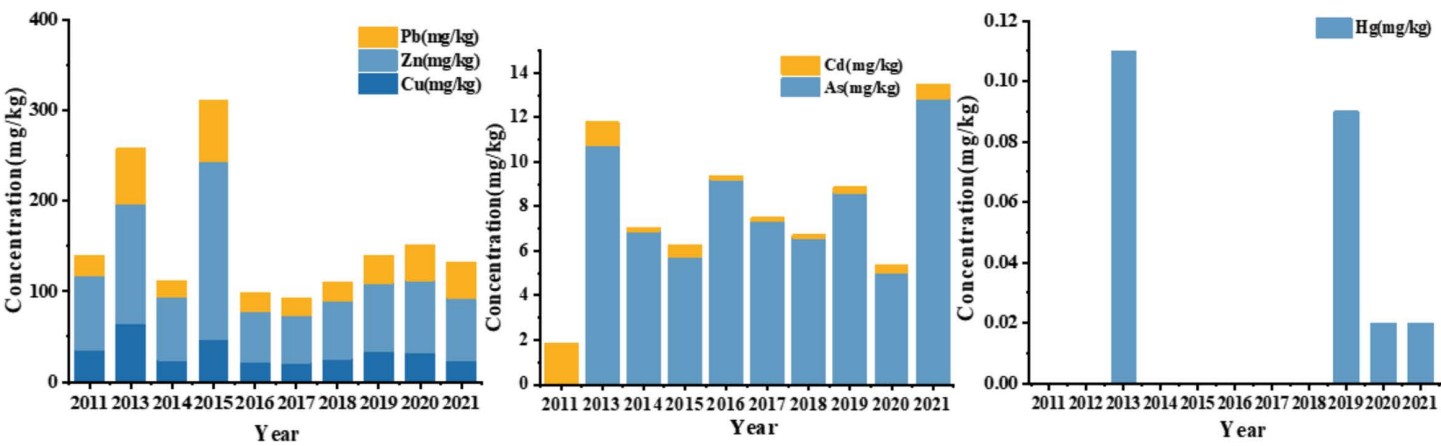

**Fig 7. Annual Variations in Heavy Metal Accumulation in Sediments.**

[19,21]. The 2015 surge (45.11 mg/kg), representing a 149% increase from 2013 levels (18.08 mg/kg), coincided with unregulated industrial expansion prior to environmental policy interventions.

Cadmium (Cd) concentrations exhibited a non-monotonic trend, decreasing from 1.65 mg/kg in 2013 to 0.15 mg/kg in 2019, followed by a 153% rebound to 0.38 mg/kg in 2021. The initial decline aligns with the implementation of agricultural non-point source controls under the Bohai Sea Environmental Protection Plan's Best Management Practices (2015–2020) [40].However, the recent resurgence underscores persistent challenges in managing phosphate fertilizer runoff, particularly during extreme weather events that induce sediment resuspension [20,31].

Mercury (Hg) concentration exhibited a dramatic 83% reduction from 0.06 mg/kg in 2019 to 0.01 mg/kg in 2020, primarily attributable to the shutdown of coal-fired power plants in Tianjin's coastal region under the Air Pollution Prevention and Control Action Plan's energy transition mandates [14]. The 2013 baseline (0.006 mg/kg) remained below China's Marine Sediment Quality Standard (GB 18668−2002) threshold values.

Arsenic (As) levels increased by 64% from 6.52 mg/kg in 2019 to 10.72 mg/kg in 2021, spatially correlating with intensive aquaculture operations in Laizhou Bay, where organoarsenical pesticides remain prevalent despite regulatory restrictions [13,15]. The 2013 baseline (0.230 mg/kg) suggested historically low contamination prior to the recent expansion of agricultural activities.

## 5 Conclusion

This study elucidates the spatial, seasonal, and interannual variations of heavy metal contamination in the Bohai Sea, revealing key anthropogenic and environmental drivers. The findings indicate that Cu, Zn and Pb concentrations peak during summer, influenced by monsoon-driven sediment resuspension and intensified industrial activities, while winter Pb surges are linked to coal combustion emissions. Interannual trends reflect regulatory impacts, with significant declines in Cu, Zn and Hg concentrations following stringent pollution control measures, whereas Cd and As contamination persists, highlighting ongoing challenges in agricultural runoff and aquaculture practices.

The study underscores the urgent need for adaptive sediment management and real-time pollution monitoring to mitigate localized contamination hotspots, particularly in industrialized bays and near major riverine inputs. By integrating hydrodynamic models with ecological risk assessments, policymakers can better predict pollutant dispersal and implement targeted interventions, such as seasonal dredging restrictions and sediment stabilization techniques.

Future research should focus on long-term metal bioavailability and trophic transfer, assessing how heavy metals accumulate in marine food webs and impact ecological health. Additionally, advancing high-resolution remote sensing and machine learning models could enhance monitoring precision and improve predictive capabilities for heavy metal pollution under evolving climate and industrial conditions. A comprehensive, multidisciplinary approach is essential to safeguard the Bohai Sea's ecological integrity and support sustainable coastal resource management.

## Supporting information

**S1 Table. Geographical information of sampling stations.**
(DOCX)

**S2 Table. Seasonalextremesandaveragesforvariousmetals.**
(DOCX)

**S3 Table. Comprehensive pollution of heavy metals in surface sediments in the central Bohai Sea.**
(DOCX)

**S4 Table. Single factor results of heavy metal pollution in sediments in different seasons.**
(DOCX)

**S5 Table. Potential ecological risk factors of heavy metals in surface sediments in different seasons.**
(DOCX)

**S6 Table. Seasonal variations of heavy metals in sediments.**
(DOCX)

**S7 Table. Annual Variations in Heavy Metal Accumulation in Sediments.**
(DOCX)

## Acknowledgments

We thank the dozens of teacher, students, and community members who participated in sample collection efforts on behalf of this project.

## Author contributions

**Data curation:** Shilin Li, Jianlei Chen.

**Formal analysis:** Shilin Li, Jianshe Zhang, Yongjiang Xu.

**Funding acquisition:** yong xu.

**Investigation:** Xuzhi Zhang, Jianshe Zhang, yong xu.

**Methodology:** Jianlei Chen, Xuzhi Zhang, Jianshe Zhang, Yongjiang Xu.

**Supervision:** Jianlei Chen, Xuzhi Zhang, Yongjiang Xu.

**Writing – original draft:** Shilin Li, yong xu.

**Writing – review & editing:** Shilin Li, yong xu.

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
