## [Decision Letter · Decision Letter 0]

PLOS ONE

Dear Dr. xu,

Thank you for submitting your manuscript to PLOS ONE. After careful consideration, we feel that it has merit but does not fully meet PLOS ONE’s publication criteria as it currently stands. Therefore, we invite you to submit a revised version of the manuscript that addresses the points raised during the review process. Furthermore, the discussion on seasonal variations in heavy metal concentrations lacks depth regarding anthropogenic influences, and both the title and abstract need to be more concise and impactful. Overall, significant revisions are needed to improve clarity, relevance, and scientific contribution before resubmission.

We look forward to receiving your revised manuscript.

Kind regards,

Barathan Balaji Prasath

Academic Editor

PLOS ONE

“The Central Public-interest Scientific Institution Basal Research Fund, CAFS (NO.2023TD53).”

6. We note that Figure 1 in your submission contain [map/satellite] images which may be copyrighted. All PLOS content is published under the Creative Commons Attribution License (CC BY 4.0), which means that the manuscript, images, and Supporting Information files will be freely available online, and any third party is permitted to access, download, copy, distribute, and use these materials in any way, even commercially, with proper attribution. For these reasons, we cannot publish previously copyrighted maps or satellite images created using proprietary data, such as Google software (Google Maps, Street View, and Earth). For more information, see our copyright guidelines: http://journals.plos.org/plosone/s/licenses-and-copyright.

Reviewers' comments:

Reviewer's Responses to Questions

**Comments to the Author**

1. Is the manuscript technically sound, and do the data support the conclusions?

Reviewer #1: Yes

Reviewer #2: Partly

2. Has the statistical analysis been performed appropriately and rigorously?

Reviewer #1: Yes

Reviewer #2: Yes

3. Have the authors made all data underlying the findings in their manuscript fully available?

Reviewer #1: Yes

Reviewer #2: Yes

4. Is the manuscript presented in an intelligible fashion and written in standard English?

Reviewer #1: Yes

Reviewer #2: Yes

Reviewer #1: Title: Distribution Characteristics and Pollution Assessment of Heavy Metals in Surface Sediments in the Central Region of the Bohai Sea

• Comment: The title is informative but could be more concise for clarity and impact. Consider removing redundant terms like "Distribution Characteristics and Pollution Assessment."

• Recommendation: Revise to something like "Heavy Metal Distribution and Ecological Risk in Surface Sediments of the Bohai Sea."

Abstract

• Comment: The abstract is comprehensive but overly detailed, especially with numerical values. This can overwhelm the reader.

• Recommendation: Summarize key findings concisely, focusing on trends, risks, and implications for environmental management.

Introduction

• Strengths: Provides a strong context, linking heavy metal pollution with ecological risks and human activities in the Bohai Sea.

• Weaknesses: The introduction is slightly repetitive (e.g., overlapping details about river inputs and sediment roles).

• Recommendation: Streamline by focusing on the knowledge gap and objectives. Move some background information to the discussion.

Methods

Study Area and Sample Collection

• Comment: The study area description is detailed but lacks clear justification for choosing the central Bohai Sea.

• Recommendation: Explicitly state why this region was selected and its importance compared to adjacent areas.

Quality Assurance

• Strengths: The QA/QC measures are thorough.

• Recommendation: Include references for the detection methods (e.g., GB standards) for better credibility.

Results

Heavy Metal Content

• Comment: Tables and figures are detailed but could be overwhelming due to excess numerical values.

• Recommendation: Highlight critical trends (e.g., exceedance rates, highest/lowest concentrations) in the text rather than listing all values.

Seasonal Variations

• Comment: Seasonal trends are well-documented, but the discussion of influencing factors could be deeper.

• Recommendation: Discuss how these variations align with known hydrodynamic and anthropogenic processes in the Bohai Sea.

Discussion

Seasonal Variation

• Strengths: Comprehensive comparisons with past studies.

• Weaknesses: Too many references to specific concentrations; the focus on ecological implications is limited.

• Recommendation: Emphasize the ecological and management implications of seasonal changes instead of re-analyzing the numbers.

Ecological Risks

• Comment: The risk assessment is solid, but the methodology explanation could be simplified.

• Recommendation: Reduce redundancy in explaining the risk indices and focus on the implications for sediment management.

Long-term Trends

• Strengths: Excellent temporal comparison with historical data.

• Weaknesses: The explanation of interannual fluctuations could be more robust.

• Recommendation: Link temporal trends to regional policy changes, industrial activities, or specific events like oil spills.

Conclusion

• Comment: The conclusion is repetitive and lacks actionable insights.

• Recommendation: Summarize the key findings in one paragraph and add specific recommendations for policymakers and future research.

Tables and Figures

• Comment: Figures are informative but could benefit from improved design (e.g., larger fonts, consistent color schemes).

• Recommendation: Ensure all figures are publication-ready with clear legends and high resolution.

References

• Comment: The references are recent and relevant but lack diversity in geographical scope.

• Recommendation: Include more global studies on heavy metal pollution for broader context.

Reviewer #2: Please see the attached comments. The work seems good however, the sampling being very old and does not justify the knowledge gap. Already a good data is available from the study area. I recommend the authors can make a review instead using existing and their own data to present status of Bohai sea.

**Do you want your identity to be public for this peer review?** For information about this choice, including consent withdrawal, please see our Privacy Policy

Reviewer #1: No

Reviewer #2: No

---

## [Author Response · Author response to Decision Letter 1]

10 May 2025

Response to Comments from Editors and Reviewers

Dear Editor/Reviewer:

We greatly acknowledge the editor and reviewer for their constructive comments and advices. Thank you for your patience and these valuable comments. It is our honor to get your help to improve our manuscript entitled “Distribution characteristics and pollution assessment of heavy metals in surface sediments in the central region of the bohai sea” with the manuscript number PONE-D-25-01587.

We have revised the manuscript according to every single comment which made by the editor and reviewers. We check the editor and reviewers’ comments repeatedly and closely to ensure each comment is addressed. The following is our reply to the comments in red color. Moreover, the changes have been highlighted in red color in a marked copy of the revised manuscript (in submitted documents).

1.Please clarify which server the basemap tiles for Figure 1 were procured from.

Response: Thank you for your comments. Regarding the basemap tiles used in Figure 1 created with Golden Software Surfer, please be informed that:

Server Source: The basemap was specifically obtained through the OpenStreetMap (OSM) tile server integrated within Golden Software Surfer's interface.

License Compliance:

OSM data is licensed under the Open Database License (ODbL) which is compatible with CC BY 4.0 for map tiles (as per OSM's guidance on map tile reuse:

https://operations.osmfoundation.org/policies/tiles/).

Open Database License (ODbL):https://opendatacommons.org/licenses/odbl/1-0/

Direct Links: https://www.openstreetmap.org/copyright

Basemap Source: https://www.openstreetmap.org/export#map=7/38.941/120.103&layers=T

Tile Usage Policy: https://operations.osmfoundation.org/policies/tiles/

We confirm that the basemap usage complies with both OSM's attribution requirements and PLOS ONE's CC BY 4.0 license policy.

2. Please clarify what program(s) were used to create this new version, and whether any external images or assets were used to create it.

Response: Thank you for your comments.The study area and sampling stations were created using Golden Software Surfer and are shown in Figure 1.No external images or assets were used.

3. Please state what role the funders took in the study.

Response:Thank you for your comments.

Funding: This study was supported by the Central Public-interest Scientific Institution Basal Research Fund, CAFS (NO.2023TD53).The funders had no role in the study design, data collection and analysis, decision to publish, or preparation of the manuscript.

4. Data availability statement.

Response:Thank you for your comments. We have provided the mean values, standard deviations, and values used to construct the graphs reported in the studies in the Supporting Information.All relevant data are within the paper and its Supporting Information files.

5. We note that Figure 1 in your submission contain [map/satellite] images which may be copyrighted.

Response:Thank you for your comments. We have revised Figure 1. Please refer to the manuscript for detailed modifications.

Response to Reviewer #1:

1.Title:Distribution Characteristics and Pollution Assessment of Heavy Metals in Surface Sediments in the Central Region of the Bohai Sea

Comment: The title is informative but could be more concise for clarity and impact. Consider removing redundant terms like "Distribution Characteristics and Pollution Assessment."

Recommendation: Revise to something like "Heavy Metal Distribution and Ecological Risk in Surface Sediments of the Bohai Sea."

Response: Thank you for your constructive suggestion. We have revised the title to ”Heavy Metal Distribution and Ecological Risk in Surface Sediments of the Bohai Sea”. Please see the manuscript for detailed modification content. (lines 1-2)

2.Abstract

Comment: The abstract is comprehensive but overly detailed, especially with numerical values. This can overwhelm the reader.

Recommendation: Summarize key findings concisely, focusing on trends, risks, and implications for environmental management.

Response: Thank you for your constructive suggestions. We have revised the abstract to focus on key findings, trends, risks and impacts on the environmental management and ensure a more concise presentation. The revisions are marked in red below. The detailed revisions have been highlighted in the manuscript (lines 21-48).

3. Introduction

Strengths: Provides a strong context, linking heavy metal pollution with ecological risks and human activities in the Bohai Sea.

Weaknesses: The introduction is slightly repetitive (e.g., overlapping details about river inputs and sediment roles).

Recommendation: Streamline by focusing on the knowledge gap and objectives. Move some background information to the discussion.

Response: Thank you for your constructive suggestion. We have revised the manuscript by (1) adding content to address knowledge gaps and research objectives (highlighted in lines 115-140), (2) deleting redundant discussions on river input, and (3) relocating relevant background information to the Discussion section (lines 356-364). All modifications are marked for your review.

4.Methods

Study Area and Sample Collection

Comment: The study area description is detailed but lacks clear justification for choosing the central Bohai Sea.

Recommendation: Explicitly state why this region was selected and its importance compared to adjacent areas.

Response: Thank you for your constructive suggestion. We have supplemented the rationale for selecting the central Bohai Rim as the study area in the Introduction (lines 128-140), explicitly stating its unique environmental significance compared to adjacent regions and the scientific justification for this geographical focus. The revised content has been highlighted for your review.

5.Quality Assurance

Strengths: The QA/QC measures are thorough.

Recommendation: Include references for the detection methods (e.g., GB standards) for better credibility.

Response: Thank you for your constructive suggestion. We have revised the manuscript by (1) incorporating references to GB 17378-2007 and methodological specifications in Section 2.2 Sample Collection and Processing (lines 170-172), and (2) expanding the QA/QC measures with detailed citations of GBW07314 (lines 180-185). All revisions are highlighted in the text for your review.

6.Results

Heavy Metal Content

Comment: Tables and figures are detailed but could be overwhelming due to excess numerical values.

Recommendation: Highlight critical trends (e.g., exceedance rates, highest/lowest concentrations) in the text rather than listing all values.

Response: Thank you for your constructive suggestion. We have revised the manuscript by incorporating a comparative chart (Fig 2) in Section 3 to visualize the minimum, maximum and average values of the six heavy metals, thereby emphasizing key trends. The original Table 3 has been streamlined accordingly, with all modifications marked in the text (lines 246-247) for your reference.

7.Seasonal Variations

Comment: Seasonal trends are well-documented, but the discussion of influencing factors could be deeper.

Recommendation: Discuss how these variations align with known hydrodynamic and anthropogenic processes in the Bohai Sea.

Response: Thank you for your constructive suggestion. We have strengthened Section 4.1 (Seasonal Variation of Heavy Metals Concentrations in the Central Bohai Sea) by (1) adding seasonal trend analysis linked to hydrodynamic and anthropogenic drivers, (2) conducting multi-factor influence evaluations, and (3) integrating these revisions into the manuscript (lines 356-397), all marked for your review.

8.Discussion

Seasonal Variation

Strengths: Comprehensive comparisons with past studies.

Weaknesses: Too many references to specific concentrations; the focus on ecological implications is limited.

Recommendation: Emphasize the ecological and management implications of seasonal changes instead of re-analyzing the numbers.

Response: Thank you for your constructive suggestion. We have enhanced the Discussion section (4.1 Seasonal Changes in Heavy Metal Concentrations in the Central Bohai Sea) by incorporating ecological and management implications strategies related to seasonal variations, with these critical additions highlighted in the manuscript (lines 398-416) for your review.

9.Ecological Risks

Comment: The risk assessment is solid, but the methodology explanation could be simplified.

Recommendation: Reduce redundancy in explaining the risk indices and focus on the implications for sediment management.

Response: Thank you for your constructive suggestion. We have simplified the explanation of the risk index and added implications for sediment management.The modification content is marked as follows in red font. Please see the manuscript for detailed modification content(lines 419-443).

10.Long-term Trends

Strengths: Excellent temporal comparison with historical data.

Weaknesses: The explanation of interannual fluctuations could be more robust.

Recommendation: Link temporal trends to regional policy changes, industrial activities, or specific events like oil spills.

Response: Thank you for your constructive suggestion. We have enhanced the explanation of interannual fluctuations by linking the temporal trends to regional policy changes, industrial activities, and specific events, such as oil spills.The modification content is marked as follows in red font. Please see the manuscript for detailed modification content(lines 459-501).

11.Conclusion

Comment: The conclusion is repetitive and lacks actionable insights.

Recommendation: Summarize the key findings in one paragraph and add specific recommendations for policymakers and future research.

Response: Thank you for your constructive suggestion. We have revised the conclusion to summarize the key findings in a more concise manner and included specific recommendations for policymakers and future research. The modification content is marked as follows in red font. Please see the manuscript for detailed modification content(lines 502-524).

12.Tables and Figures

Comment: Figures are informative but could benefit from improved design (e.g., larger fonts, consistent color schemes).

Recommendation: Ensure all figures are publication-ready with clear legends and high resolution.

Response: Thank you for your constructive suggestion. We have modified all the figures, including making the font larger, the color scheme consistent, and the legend clearer, to ensure that the figures can be published and have high resolution. The modified figures are as follows. Please see the manuscript for detailed changes.

13 References

Comment: The references are recent and relevant but lack diversity in geographical scope.

Recommendation: Include more global studies on heavy metal pollution for broader context.

Response: Thank you for your constructive suggestion. We have added relevant content on global heavy metal pollution research in the introduction, including (Mediterranean Sea, Persian Gulf, Red Sea-Gulf of Aqaba coast and the southern Gulf of Mexico)The modification content is marked as follows in red font.Please see the manuscript for detailed modification content.(lines 115-127).And adjusted have the format of references according to the journal standards.

Response to Reviewer #2:

1.The Manuscript assesses surface sediment contamination in Bohai sea region. The authors has put in good efforts for such extensive sampling.

Response: Thank you for your affirmation of our work.

2. As the area is highly researched already by several researchers and hence, the present work do not present a novel or knowledge gap fulfilment. Line 105-108: There is already much research done on heavy metal conc. In Bohai sea. Already many of them included in literature.

Response: Thank you for your comments. Yes, this study does not fill the knowledge gap. We have revised the manuscript. The existing literature focuses on extensive research on the area around Bohai Sea. However, we found only a few studies on the central area of Bohai Sea. Since the 2011 oil spill caused pollution in Bohai Sea, we hope that by comparing the environmental changes ten years ago with the present, our findings aim to inform future environmental management strategies in this critical marine ecosystem.

3. The sediment samples collection is already a decade old. Also, is the present analysis using AAS is done presently or during 2013 which needs to be made clear.

Response: Thank you for your comments. The sediment sample collection and the Atomic Absorption Spectroscopy (AAS) analysis were both conducted in 2013. We have clarified this in the revised manuscript to avoid any confusion. The modification content is marked as follows in red font (2.2 Sample collection and processing). Please see the article for detailed modification content (lines172).

4. Line 125 – capitalize

Response: Thank you for your careful review and for pointing out the formatting issue. We have capitalized the letters in line 125.Please see the article for detailed modification content (lines150).

We appreciate your attention to detail and will ensure that the manuscript adheres to the proper formatting standards.

5. As the overall research is carried out on the base of 2013 year sampling. I think the data is a more than a decade old and to put forward it in present scenario will have less importance in present context.

Response: Thank you for your comments on the timeliness of the data. We understand your concerns, especially about the rapid changes that may have occurred in the environment over the past decade. Since pollution in the central Bohai Rim is a long-term problem, we used data from 2013 because recent studies Li et al. (2020) and Wei et al.(2022) examined data from the central Bohai Rim. The comparison provides an understanding of environmental changes in the central Bohai Sea after the 2011 oil spill that caused pollution in the Bohai Rim region. The patterns observed in 2013 provide a basis for understanding broader trends and the historical context of pollution.

e.g:

[1] Li LQ, Hu H, Lv XL, Liu Q. Contents and ecological risk assessment of heavy metals in surface sediments of the Central Bohai Sea and the Northern Yellow Sea. Trans Oceanol Limnol. 2020;1: 84-92.

[2] Wei Y, Ding D, Qu K, Sun J, Cui Z. Ecological risk assessment of heavy metal pollutants and total petroleum hydrocarbons in sediments of the Bohai Sea, China. Mar Pollut Bull. 2022;184: 114218.

6. Moreover, already good quantity of work is already been undertaken in the study area hence the study does not fill the knowledge gap.

Response: Thank you for your comments. Yes, this study does not fill the knowledge gap. We have revised the manuscript. Since the 2011 oil spill caused pollution in Bohai Sea, we hope that by comparing the environmental changes ten years ago with the present, it may have an impact on future environmental management strategies.

7.As already mentioned in introduction and discussion section, the study area is a highly polluted area, which the result presented the moderate to low quantity of pollution.

Response: Thank you for your comments. This difference is due to the different research scopes. In the introduction and discussion, the comparative research scope is the heavily polluted area of the Bohai oil spill accident, while this study is the central area of the Bohai Sea. After the Penglai 19-3 oil spill accident, the Chinese government quickly took pollution blocking measures to shorten the polluted area as much as possible, so the sampling points failed to fully cover the heavily polluted area, so the pollution level was medium to low.

8.The seasonal changes in the heavy metal concentration are not properly justifiable. Specially, the study mentions a lot anthropogenic activities in the bay region. Although, the sources and activities are not well discussed.

Response: Thank you for your comments. We have supplemented the discussion of human activities in the Bohai Sea Rim region in the revised manuscript 4.1 (Seasonal Variation of Heavy Metal Concentrations in the Central Bohai Sea) and provided a more detailed explanation of the potential sources of heavy metal pollution, including in

---

## [Decision Letter · Decision Letter 1]

Heavy Metal Distribution and Ecological Risk in Surface Sediments of the Bohai Sea

PONE-D-25-01587R1

Dear Dr. Yong Xu,

We’re pleased to inform you that your manuscript has been judged scientifically suitable for publication and will be formally accepted for publication once it meets all outstanding technical requirements.

Kind regards,

Barathan Balaji Prasath

Academic Editor

PLOS ONE

Additional Editor Comments (optional):

Reviewers' comments:

Reviewer's Responses to Questions

**Comments to the Author**

Reviewer #1: (No Response)

Reviewer #2: All comments have been addressed

2. Is the manuscript technically sound, and do the data support the conclusions?

Reviewer #1: Yes

Reviewer #2: Yes

3. Has the statistical analysis been performed appropriately and rigorously?

Reviewer #1: Yes

Reviewer #2: Yes

4. Have the authors made all data underlying the findings in their manuscript fully available?

Reviewer #1: Yes

Reviewer #2: Yes

5. Is the manuscript presented in an intelligible fashion and written in standard English?

Reviewer #1: Yes

Reviewer #2: Yes

Reviewer #1: No Comments. I have fully satisfied with current format and no conflict of interest statement with this paper.

Reviewer #2: It seems that most of the reviewers comments are well addressed and justified. The MS can be accepted with final editorial decision.

**Do you want your identity to be public for this peer review?** For information about this choice, including consent withdrawal, please see our Privacy Policy

Reviewer #1: **Yes: ** Dr. Nikunj B. Gajera

Reviewer #2: No

---

## [Editor Report · Acceptance letter]

PONE-D-25-01587R1

PLOS ONE

Dear Dr. xu,

I'm pleased to inform you that your manuscript has been deemed suitable for publication in PLOS ONE. Congratulations! Your manuscript is now being handed over to our production team.

Kind regards,

on behalf of

Dr. Barathan Balaji Prasath

Academic Editor

PLOS ONE